# Glycolysis in Chronic Liver Diseases: Mechanistic Insights and Therapeutic Opportunities

**DOI:** 10.3390/cells12151930

**Published:** 2023-07-26

**Authors:** Hengdong Qu, Junli Liu, Di Zhang, Ruoyan Xie, Lijuan Wang, Jian Hong

**Affiliations:** Department of Pathophysiology, School of Medicine, Jinan University, Guangzhou 510632, China; quhengdong@stu2021.jnu.edu.cn (H.Q.);

**Keywords:** chronic liver disease, glycolysis, HIF-1α, metabolic reprogramming, immune activation, PKM2, therapeutic target

## Abstract

Chronic liver diseases (CLDs) cover a spectrum of liver diseases, ranging from nonalcoholic fatty liver disease to liver cancer, representing a growing epidemic worldwide with high unmet medical needs. Glycolysis is a conservative and rigorous process that converts glucose into pyruvate and sustains cells with the energy and intermediate products required for diverse biological activities. However, abnormalities in glycolytic flux during CLD development accelerate the disease progression. Aerobic glycolysis is a hallmark of liver cancer and is responsible for a broad range of oncogenic functions including proliferation, invasion, metastasis, angiogenesis, immune escape, and drug resistance. Recently, the non-neoplastic role of aerobic glycolysis in immune activation and inflammatory disorders, especially CLD, has attracted increasing attention. Several key mediators of aerobic glycolysis, including HIF-1α and pyruvate kinase M2 (PKM2), are upregulated during steatohepatitis and liver fibrosis. The pharmacological inhibition or ablation of PKM2 effectively attenuates hepatic inflammation and CLD progression. In this review, we particularly focused on the glycolytic and non-glycolytic roles of PKM2 in the progression of CLD, highlighting the translational potential of a glycolysis-centric therapeutic approach in combating CLD.

## 1. Introduction

Chronic liver disease (CLD) encompasses a broad spectrum of liver diseases, ranging from viral hepatitis, alcoholic liver disease (ALD), and nonalcoholic fatty liver disease (NAFLD) to end-stage hepatic conditions including nonalcoholic steatohepatitis (NASH), liver fibrosis, hepatocellular carcinoma (HCC), and cholangiocarcinoma (CCA) [1]. Due to the growing prevalence of metabolic syndromes, including obesity and hyperlipidemia, NAFLD has become the most prevalent CLD, affecting more than 25% of the adult population worldwide [2]. Currently, there are no approved therapeutics for certain CLDs, such as NASH and liver fibrosis, which is partly limited by the lack of understanding of their complicated pathogenesis and druggable targets [3,4].

Liver is the most vital organ for glucose homeostasis, where it generates glucose during fasting and reserves glucose postprandially [5,6]. During the progression of CLD, hepatocyte injury and chronic low-grade inflammation lead to metabolic dysfunction, which causes an excessive accumulation of lipids and aberrant activation of metabolic pathways in the liver, including enhanced aerobic glycolysis [7,8]. On one hand, aerobic glycolysis promotes the pro-inflammatory activation of immune cells, which perpetuates hepatic inflammation and liver injury [9]. On the other hand, HCC encompasses enhanced aerobic glycolysis to support the proliferation, metastasis, and drug resistance of HCC cells [10]. Growing evidence has highlighted the significance of aerobic glycolysis in the progression of CLDs, suggesting that targeting abnormal glycolytic flux may serve as an effective strategy to combat CLDs [11,12,13,14].

Several glycolytic mediators have been reported to regulate the progression of CLDs. Pyruvate kinase (PK) is a rate-limiting enzyme that catalyzes the final step of glycolysis. It converts phosphoenolpyruvate (PEP) into pyruvate and supports ATP production during glycolysis. Unlike mitochondrial respiration, PK produces ATP independent of oxygen supply, which allows organs to survive under hypoxic conditions [15]. Due to the unique metabolic requirements of tissues, the expression levels of each pyruvate kinase isozyme vary substantially in both kinetics and regulatory mechanisms. There are four subtypes of PKs, ranging from PKL and PKR encoded by *PKLR* gene to PKM1 and pyruvate kinase M2 (PKM2) encoded by the *PKM* gene [16]. Unlike other isoforms that exclusively function as hyperactive tetramers and promote metabolic flux toward oxidative phosphorylation (OXPHOS), PKM2 contains a less active monomeric and dimeric form, which shifts the metabolite from OXPHOS to aerobic glycolysis [17]. In addition, the PKM2 dimer can translocate to the nucleus and act as a transcriptional coactivator to regulate gene expression [18]. Interestingly, PKM2 directly binds to HIF-1α and promotes the transcription of a series of glycolytic genes, including PKM2, thereby forming a positive feedback loop for aerobic glycolysis (Figure 1) [19].

Owing to its unique properties, PKM2 is preferentially upregulated in immune and cancerous cells, which display high levels of nucleic acid synthesis [20]. Previous works on PKM2 have mainly investigated its effect on the metabolism, proliferation, and migration of tumor cells [21]. Recent studies have shown that PKM2 is involved in immune activation via reprogramming glycolysis [22,23]. In this review, we summarize the neoplastic and non-neoplastic role of aerobic glycolysis in the progression of CLD by particularly focusing on PKM2, highlighting the translational potential of applying PKM2 as a druggable target to combat CLDs.

## 2. The Expression of PKM2 in CLDs

In line with the dynamic metabolic conditions and characteristics of different tissues, levels of PK isoforms are highly regulatory and tissue-specific. PKL and PKR are primarily expressed in the healthy liver, intestine, and red blood cells. PKM1 is expressed in adult tissues, including the bone and brain, whereas PKM2 is expressed in embryonic cells, undifferentiated tissues, and tumors [21]. Consistent with this notion, PKM2 expression is hardly detectable in healthy livers but is dramatically upregulated in liver cancer [24,25]. Interestingly, although a switch from PKM1 to PKM2 regulated by the alternative splicing of PKM was observed in many types of tumor, PKM2 was the prominent isoform of PKM both in normal liver and HCC [26]. During DEN/CCl_4_- or STZ/HFD-induced murine hepatocarcinogenesis, the level of PKM2 in the plasma ectosome gradually increased before tumor formation. Moreover, PKM2 was significantly enriched in ectosomes from patients with HCC compared to healthy donors, indicating that PKM2 may serve as an early diagnostic marker for HCC [27]. In patients with HCC, PKM2 is highly upregulated in tumor tissues and is positively correlated with poor prognosis [28,29,30]. Furthermore, the overexpression of PKM2 in HCC is correlated with a high TNM stage and level of vascular invasion, and patients who are positive for PKM2 expression have an increased incidence of postsurgical HCC recurrence [31,32].

Liver cirrhosis, the progressive stage of liver fibrosis, is recognized as a key mediator in the pathogenesis of liver cancer as it progresses to HCC in up to 90% of patients [33]. Interestingly, PKM2 expression is significantly increased in precancerous cirrhotic livers and strongly associated with an elevated risk of developing HCC [34]. Moreover, the level of hepatic PKM2 is higher in cirrhotic HCC than in non-cirrhotic HCC, suggesting that PKM2 plays an important role in the regulation of the precancerous and tumor microenvironments of HCC [35]. The expansion of PD-L1^+^ tumor-associated macrophages is a critical driver of the immune escape of HCC and correlates with poor prognosis in HCC patients [36]. Notably, PKM2 is overexpressed in PD-L1^+^ glycolytic macrophages, and PD-L1 blockade liberates the intrinsic antitumorigenic properties of PKM2^+^ macrophages, indicating that PKM2 may serve as an indicator for precise anticancer therapy [37]. PKM2 is also upregulated in CCA tissues compared to healthy liver tissues and is positively correlated with the poor prognosis of CCA patients [38]. Serum PKM2 levels are elevated in patients with precancerous cholangitis, and CCA and can be used to discriminate between benign and malignant stages of disease with high specificity and sensitivity [39]. These studies suggest that PKM2 is a key player in the progression of liver cancer and may serve as an effective prognostic and predictive biomarker.

In addition to cancerous conditions, PKM2-mediated aerobic glycolysis plays a critical role in inflammatory disorders and nonneoplastic liver diseases [19,23]. The expression of PKM2 is upregulated in HSC in the context of liver fibrosis and is overexpressed in hepatic macrophages and Th17 cells during NAFLD/NASH development [40,41,42]. Of note, serum and hepatic levels of PKM2 are increased in both metabolic syndrome and NAFLD, but they decreased after Roux-en-Y gastric bypass surgery, one of the most commonly performed weight-loss procedures, implying that systematic PKM2 expression may represent the disease stage of NAFLD [43]. Currently, there are no effective noninvasive diagnostic methods for NAFLD, NASH, or liver fibrosis. The pattern of PKM2 expression in inflammatory liver diseases may lead to the development of novel biomarkers for accurate diagnosis, either independently or along with liver biopsy, which is the gold standard for NASH diagnosis [44,45]. Collectively, it would be of great clinical significance to characterize the expression of PKM2 in CLDs for the development of novel diagnostic and prognostic biomarker.

## 3. Current Status and Challenge of Liver Cancer

The liver is the sixth most prevalent site of primary cancers, including HCC and CCA. Liver cancer is the fourth leading cause of cancer-related deaths worldwide owing to its high incidence of postsurgical recurrence and metastasis [46]. In particular, HCC accounts for 85–90% of liver cancers [47]. Owing to the increased global incidence of metabolic syndrome, NAFLD has become the most prominent cause and risk factor for HCC in numerous developed countries [48]. Traditionally, systemic therapies, including sorafenib or lenvatinib, have been practiced as the first-line therapy. Recently, ICIs have revolutionized HCC treatment, with a significant increase in patient survival [49]. A combination of Atezolizumab with Bevacizumab was approved as first-line HCC therapy in 2020. Tremelimumab and durvalumab were also approved as first-line HCC therapies in 2022. Despite these major advances, NAFLD has been reported to limit the therapeutic efficacy of ICI in treating HCC, and a better stratification system based on different individuals is still needed to guide clinical decision making [50].

### 3.1. PKM2 in HCC

Aerobic glycolysis is one of the most prominent features of liver cancer which supports a broad range of oncogenic regulation, including proliferation, metastasis, immunosuppression, and drug resistance [7]. In this section, we introduce recent advances in PKM2-directed glycolysis for HCC formation and progression. Several mechanisms have been reported to modulate PKM2-mediated aerobic glycolysis and the progression of HCC. HSP90 promotes the Warburg effect and proliferation of HCC cells via direct binding to PKM2 and phosphorylates it at Thr-328, which is a site that is critical for sustaining PKM2 stability [51]. Circular RNA MAT2B sponges miR-338-3p and promotes the expression of PKM2, thereby enhancing aerobic glycolysis and HCC progression under hypoxia [52]. Yu et al. found that MTR4, an RNA helicase, drives cancer metabolism and HCC progression by ensuring the alternative splicing of specific glycolytic genes, including PKM2 [53]. HIF-1α plays a critical role in regulating the transcription of glycolytic genes, especially in tumor microenvironments, including HCC [54,55]. Under hypoxia, YAP binds to HIF-1α in the nucleus, which thereby maintains HIF-1α stability and the aerobic glycolysis of HCC. Moreover, HIF-1α binds to *PKM* mRNA and directly activates the transcription of PKM2, accelerating the glycolysis of HCC cells [56]. Meanwhile, PKM2 is known to regulate HIF-1α transactivation, which results in an upregulation of several glycolytic genes, including LDHA and PKM2 [57,58]. This positive feedback loop may further fuel aerobic glycolysis and cause drug resistance to PKM2-targeting therapy [59].

The nuclear translocation of PKM2 is considered an indispensable course in the stimulation of aerobic glycolysis, progression, and drug resistance in HCC. Enhanced aerobic glycolysis is associated with HCC resistance to sorafenib, whereas the disruption of PKM2-associated glycolysis increases apoptosis and re-sensitizes resistant tumor cells to sorafenib [60]. A study showed that PRMT6 promotes PKM2 nuclear translocation, leading to increased aerobic glycolysis in HCC, while the addition of 2-DG (a well-known inhibitor of glycolysis) sufficiently reverses PRMT6 deficiency-mediated tumor progression and sorafenib resistance [61]. Zhou et al. reported that GTPBP4 induces PKM2 sumoylation and dimer formation. Dimeric PKM2 further translocates into the nucleus, thereby facilitating EMT and aerobic glycolysis in HCC via the STAT3 signaling pathway [62]. Myofibroblasts MyD88-mediated CCL20 secretion promoted PKM2 nuclear translocation and aerobic glycolysis in HCC cells via an ERK-dependent signaling pathway [63]. Additionally, PKM2 has been reported to fuel HCC metastasis and inhibit autophagy through the JAK/STAT3 pathway [64].

PKM2 also contributes to the development of an immunosuppressive microenvironment during HCC progression [65]. PKM2 levels were positively correlated with the levels of immune inhibitory cytokine and immune cell infiltration in HCC [28]. Lu et al. reported that PD-L1^+^ macrophages display high levels of glycolysis via the PKM2/HIF-1α axis triggered by fibronectin 1 derived from HCC cells [37]. Extracellular vesicles derived from tumor cells are critical mediators of cell-to-cell communication in the setting of tumorigenesis [66]. Ectosome PKM2 released by HCC cells facilitates monocyte-to-M2 macrophage differentiation via the STAT3 signaling pathway and remodels an immunosuppressive microenvironment, allowing immune escape and tumor progression [27]. Although PKM2 exhibits a dramatic promoting effect on HCC progression, the global ablation of PKM2 results in spontaneous tumor formation, highlighting the complexity of PKM2 in regulating HCC [67].

### 3.2. PKM2 in CCA

CCA is a highly lethal adenocarcinoma of the hepatobiliary system that is characterized by late diagnosis, short-term survival, and chemoresistance [68]. PKM2-associated aerobic glycolysis is also enhanced in CCA cells, resulting in low levels of pyruvate, a decreased inhibitory effect on HDAC3, and the suppression of apoptosis [69]. Furthermore, PKM2 is recognized as a key player in regulating EMT in CCA [70]. The knockdown of PKM2 effectively inhibits the migration, invasion, and proangiogenic capability of CCA cells via the downregulation of EMT-related markers [71]. Yu et al. provided in vivo evidence that PKM2 inhibition suppresses CCA cell proliferation, tumor growth, and neural invasion [38]. Moreover, the overexpression of CNRIP1 (a tumor suppressor) facilitated PKM2 degradation by activating parkin, which inhibited CCA progression in a mouse xenograft model [72]. These findings accentuate the potential of targeting PKM2 to combat CCA.

## 4. Inflammatory Liver Diseases

The liver is generally considered a vital organ that participates in metabolism, nutrient storage, and detoxification. During these complex processes, the hepatic immune system is challenged by numerous bacterial stimuli and harmful molecules. Maintaining homeostasis requires the liver to be immunotolerant while remaining alert to potential infectious agents or tissue damage [73]. Owing to these unique characteristics, the mechanisms that resolve hepatic inflammation are extremely important [74]. Failure to sustain tissue homeostasis leads to inflammation and liver injury, potentiating the development of fibrosis, cirrhosis, and even HCC.

### 4.1. PKM2 in Fatty Liver Diseases

Liver steatosis, which is attributed to obesity, alcohol use, or chemical-induced injury, may lead to fatty liver disease and further progress to steatohepatitis in the presence of inflammation [75]. During this process, M1 macrophages exacerbate hepatic inflammation and disease progression, whereas M2 macrophages protect against steatosis and liver fibrosis [76]. Particularly, the PKM2-driven progression of fatty liver disease is mainly dependent on metabolic reprogramming and the M1 polarization of hepatic macrophages (Figure 2). PKM2-mediated glycolysis is enhanced during macrophage M1 polarization in NASH, which correlates with miR-122-5p downregulation [40]. Kong found that HSPA12A binds to PKM2 and stimulates its nuclear translocation, which further provokes macrophage M1 polarization and the secretion of pro-inflammatory cytokines, including IL-1β and IL-6, ultimately leading to hepatocyte steatosis via paracrine effects [77]. PKM2 is also upregulated in hepatocytes during steatosis, and the disruption of PKM2 activity alleviates mitochondrial ROS and hepatocyte lipid accumulation [24]. Moreover, PKM2 has been shown to regulate the metabolic skewing of Th17 cells, and cell-specific PKM2 knockout effectively ameliorates hepatic inflammation and NAFLD [41]. In ALD, hepatocyte DRAM1 is upregulated in response to excessive ethanol, which increases PKM2-enriched extracellular vesicles, thereby promoting macrophage M1 activation and hepatic inflammation [78].

### 4.2. PKM2 in Liver Fibrosis and Cirrhosis

PKM2 is involved in the progression of liver fibrosis, which is a major cause of mortality in patients with end-stage liver disease, and it is characterized by hepatocyte injury and HSC (hepatic stellate cell) activation. Macrophages play an important role in perpetuating hepatic inflammation and HSC activation via the release of pro-inflammatory cytokines [79]. Rao et al. found that FSTL1 promotes PKM2 stability and nuclear translocation in macrophages, which further enhances macrophage M1 polarization, the production of pro-inflammatory cytokines, HSC activation, and liver fibrosis [80]. PKM2 in HSC also promotes its activation and fibrogenesis by facilitating aerobic glycolysis by regulating histone H3K9 acetylation in activated HSCs [42]. Interestingly, activated HSC can release PKM2-enriched exosomes that induce the glycolysis and activation of quiescent HSCs, hepatic macrophages, and LSECs, forming a positive feedback loop that promotes the progression of liver fibrosis [81].

## 5. Therapeutic Opportunities of PKM2-Targeted Therapy 

PKM2-targeting molecules have been mainly characterized as inhibitors and agonists (Table 1). When inhibitors limit PKM2 tetramer formation, agonists induce the transformation of PKM2 dimers into tetramers, thereby limiting its nuclear translocation [82,83]. Although both inhibitors and agonists can inhibit PKM2-mediated glycolysis and immune activation, whether an inhibitor could affect PKM2 nuclear translocation remains incompletely understood [58,59,84,85,86]. Nevertheless, treatments including traditional Chinese medicine or nucleotides-related therapeutics have been shown to modulate PKM2 activity in CLDs. In this section, we highlight the translational potential of PKM2-targeting therapy in combating CLDs. 

In neoplastic liver diseases, the inhibition of PKM2 by either shikonin or compound 3k suppresses glycolysis and proliferation, induces apoptosis in HCC cells in vitro and enhances the antitumor effect of sorafenib in vivo [87,88,98]. Similarly, shikonin has been reported to inhibit the growth and migration of CCA cells in vitro, whereas the in vivo evidence remains lacking [90,91]. Meanwhile, shikonin aggravates the oxidative stress and nutrient deficiency of HCC cells by causing mitochondria dysfunction, which further validates the efficacy of PKM2 inhibition in treating HCC [99]. Transarterial chemoembolization (TACE) is a palliative and neoadjuvant treatment for HCC patients [100]. The upregulation of PKM2 is strongly associated with a decreased response rate and shortened survival in patients receiving TACE, whereas the inhibition of PKM2 by shikonin effectively improves the efficacy of TACE in resistant cells [101]. Notably, Lu et al. reported that shikonin unexpectedly induced PKM2 nuclear translocation and the transcription of BAG3, a gene related to sustained cell survival, suggesting that a combination of a BAG3 inhibitor and shikonin may exhibit better anti-HCC efficacy [85]. The PKM2 activator can also be used to treat HCC. Unlike inhibitors, PKM2 activators display antitumor effects by enhancing pyruvate kinase activity, resulting in complete glycolysis and decreased anabolism, thereby inhibiting the growth of solid tumors including HCC [102,103]. In addition to the PKM2 inhibitor and activator, protein hydrolysate extracted from Oviductus Ranae reduces PKM2 expression by upregulating miR-491-5p and thereby efficiently prohibited HCC growth and metastasis [104]. Moreover, PKM2 shifts metabolites to aerobic glycolysis, whereas PKM1 drives metabolism toward oxidative phosphorylation. An antisense oligonucleotide (ASO) that switches *PKM* splicing from tumor-promoting PKM2 to the PKM1 isoform limits aerobic glycolysis, thereby inhibiting HCC growth both in vitro and in vivo, laying the groundwork for a potential ASO-based splicing therapy in treating liver cancer [105,106].

PKM2 is also a promising target for the treatment of inflammatory CLDs. Gwon et al. discovered that shikonin attenuates HFD-induced NAFLD by stimulating fatty acid oxidation and energy expenditure via AMPK activation [92]. Tong et al. found that shikonin can alleviate liver fibrosis by downregulating the TGF-β1/Smad pathway [93]. Although the role of PKM2 was not emphasized in the above studies, PKM2 is closely related to mitochondrial fitness and autophagy [107]. Therefore, the therapeutic efficacy of shikonin in NAFLD and liver fibrosis may be partially attributed to alterations in PKM2 activity. Furthermore, pharmacological PKM2 agonists, which limit PKM2 nuclear translocation, effectively ameliorate MCD-induced NASH in mice by re-educating macrophages from M1 to M2 polarization [95]. Annexin A5 attenuates HFD-induced NASH by regulating hepatic macrophage polarization by directly blocking PKM2 Y105 phosphorylation and nuclear translocation [108]. Digoxin, a cardiac glycoside, ameliorates steatohepatitis by disrupting PKM2–HIF-1α transactivation, thereby inhibiting metabolic reprogramming and the pro-inflammatory activation of macrophages [96]. A plant-derived triterpene celastrol that limits glycolysis and reprograms macrophage polarization from the pro-inflammatory M1 phenotype to the anti-inflammatory M2 phenotype was found to simultaneously restrain PKM2 nuclear translocation and enzymatic activity at the same time and protect against NAFLD [109]. A recent study also demonstrated that lapachol ameliorates NAFLD progression by directly inhibiting PKM2 phosphorylation and nuclear translocation, which then suppresses Kupffer cell M1 polarization [110]. Furthermore, PKM2 is involved in HSC activation, and limiting PKM2 nuclear translocation by TEPP-46 effectively attenuates the progression of liver fibrosis by inhibiting HSC activation [42,94]. These studies feature PKM2 as an attractive pharmacological target in treating CLDs. 

## 6. Conclusions and Future Perspectives

Chronic liver diseases (CLDs) encompass a broad spectrum of liver diseases ranging from ALD and NAFLD to life-threatening NASH, cirrhosis, and even HCC. The incidence of most CLDs is continuously rising when effective or approved treatments are lacking. The Warburg effect (aerobic glycolysis) plays an important role in the progression of CLDs [111,112]. In neoplastic CLDs, including HCC and CCA, the Warburg effect fuels and sustains tumor growth, metastasis, recurrence, and drug resistance [113]. In non-neoplastic CLDs, the Warburg effect is tightly linked to immune activation and hepatic inflammation, which is a condition that is profoundly involved in NASH and liver fibrosis [114,115,116]. Herein, understanding the mechanisms governing the Warburg effect in CLDs may help identify novel therapeutic targets.

PKM2 is a rate-limiting enzyme in glycolysis. Owing to its unique dimeric form with low pyruvate kinase activity, the upregulation of PKM2 is a hallmark of cells with increased aerobic glycolysis. Although numerous studies have attempted to elucidate the importance of PKM2 in the development of neoplastic diseases, its role of PKM2 in inflammatory disorders, especially CLDs, has not been fully elucidated. The translation of this knowledge into clinical practice is at a nascent stage, partly owing to the lack of studies assessing the cell-specific role of PKM2 in CLDs, as the function of PKM2 in certain hepatic cells, including LSECs and bile duct cells, remains elusive. Moreover, most studies that have investigated the role of PKM2 were based on cellular and animal models, which leaves the question of whether targeting PKM2 in human diseases will bring up beneficial effects similar to what has been observed in in vivo models. In silico models, including quantitative systems pharmacology (QSP) models, have been extensively applied to drug discovery by illustrating the molecular interactions between biological systems and drug candidates [117,118]. In particular, several well-established QSP models can be used to study glucose metabolism [119,120], the Warburg effect [121] and liver function [122], all of which may help further the translation study of PKM2 in combating CLDs.

Undoubtedly, as a therapeutic target, PKM2 has unique advantages, since its expression is almost undetectable in the healthy liver and starts to increase as the disease progresses. Furthermore, when cells undergo abnormal activation, PKM2 is mainly localized to the nucleus, potentiating the application of PKM2 activators in treating CLDs without affecting normal or quiescent cells. In conclusion, although future studies are required to illustrate the clinical significance of PKM2 targeting molecules along with their immediate and long-term health effects, PKM2 may serve as a novel therapeutic target for both neoplastic and inflammatory CLDs.

## Figures and Tables

**Figure 1 cells-12-01930-f001:**
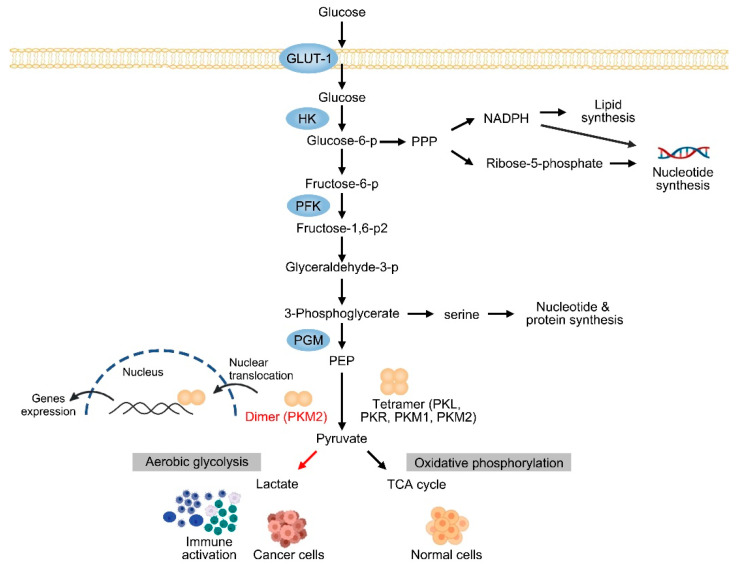
Pyruvate kinase isoforms in metabolic reprogramming.

**Figure 2 cells-12-01930-f002:**
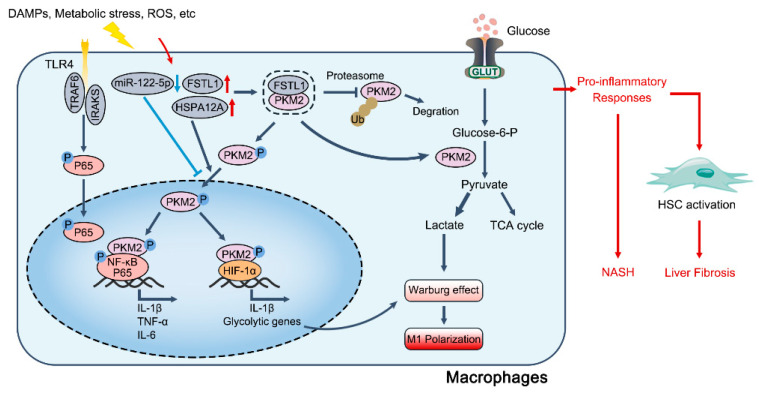
Role of macrophage PKM2 in the progression of NAFLD and liver fibrosis. During the pathogenesis of NASH and liver fibrosis, several stimuli released by injured hepatocytes activate hepatic macrophages via TLR4/NF-κB signaling. Meanwhile, levels of FSTL1 and HSPA12A are elevated when the expression of miR-122-5p is downregulated in response to liver inflammation, both of which are reported to regulate PKM2 Y105 phosphorylation and nuclear translocation. Specifically, FSTL1 directly binds to PKM2 and maintains its stability, thereby promoting PKM2-mediated glycolysis and dimer activity in M1 macrophages. On one hand, the nuclear translocation of PKM2 activates the NF-κB-directed and HIF-1α-directed transcription of pro-inflammatory genes including IL-1β. On the other hand, PKM2-HIF-1α transactivation upregulates the expression of several glycolytic genes, which further fuel aerobic glycolysis and macrophage M1 polarization. Ultimately, PKM2-mediated pro-inflammatory responses perpetuate hepatic inflammation and exacerbate the development of NASH and liver fibrosis.

**Table 1 cells-12-01930-t001:** Main characteristics of PKM2-targeting compounds as CLD therapy.

Name	Structure	MW	Disease Types	Pharmacological Properties	Refs.
C3k	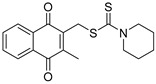	345.5	HCC	Inhibitor	[87,88]
Shikonin	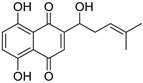	288.3	HCC	Inhibitor	[59,89]
			CCA		[90,91]
			NAFLD		[92]
			LF		[93]
ML265	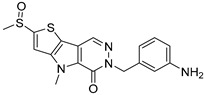	372.5	LF	Agonist	[42,94]
			NASH		[95]
DASA-58	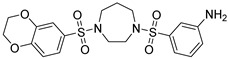	453.5	LF	Agonist	[80]
Digoxin	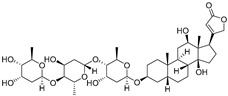	780.9	NASH	Inhibiting PKM2 trans-activation	[24,96]
Meformin	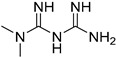	315.8	HCC	Suppressing PKM2 activity	[97]
			CCA		[38]

## Data Availability

Not applicable.

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
