# Peer review of "Glycolysis in Chronic Liver Diseases: Mechanistic Insights and Therapeutic Opportunities"

_cells, 2023, doi:10.3390/cells12151930_

Round 1
Reviewer 1 Report
This review is well organized and provides a comprehensive overview of PKM2 in chronic liver diseases (CLD) including NAFLD, NASH, HCC and CCA. The authors have described PKM2 mainly functions in non parenchymal cells to participate in the pathogenesis of liver cancer, inflammatory liver diseases, as well as the therapeutic application of PKM2 agonists and inhibitors in treating these CLD. PKM2 does not express in healthy liver but shown increased expression in HCC cells, HSC and macrophages, how PKM2 expression is regulated under normal and different CLD condition and how does its level compared with other forms of PKs in primary hepatocyte, HSC and macrophages under theses conditions? How does the three conformation ? These are also important questions the author should consider in this review. One minor point is the reference should be included by the end of a sentence, not the beginning of a sentence.
Overall, the quality of english language in this review is great.
Author Response
We thank the reviewer for these very important points and helpful suggestions. To better characterize the difference between PKM2 and other pyruvate kinase isoforms, we added a Figure in the revised manuscript. In addition, we added more illustration on the expression pattern of PKM2 and PKM1 under normal hepatic conditions (line 116-119).

Reviewer 2 Report
This is a well written comprehensive manuscript on review of PKM2 in chronic liver disease and has discussed the mechanistic details and current status of PKM2 as potential target. I would recommend accepting this manuscript with minor review to incorporate the following for holistic understanding and scope of readers. My comments are as follows:
- Expand HSC at line 230
- Since there is only corresponding author, it is recommended to add this statement "Author to whom correspondence should be addressed" at line 6
- All the references should follow the journal guidelines and should be enclosed before the end of the statement. Similar format should be followed in addressing the figure. i.e., the reference number should follow a period.
- Expand TACE at line 257
- At line 276, the statement should correct/address as Gwon SY et al.
- Apart from sorafenib, there are two other options for first-line treatment that was recently approved : Atezolizumab with Bevacizumab (approved in 2020) and Tremelimumab along with Durvalumab (approved in 2022). Authors should mention this in section 3.
- Since this manuscript is about mechanistic insights I strongly recommend to include a section on how in-silico models can assist in implementing PKM2 as a druggable target. Mechanistic details discussed in this manuscript can be used in the formulation of quantitative systems pharmacology models (QSP) such as [1],[2]. Recently QSP models have been extensively used in drug discovery and development phase. Authors should also mention some of the other well known liver models in literature such as [3-6] that can help in designinga qsp model for liver for targeting PKM2 to combat CLD.
[1] Watkins, Paul B. "DILIsym: Quantitative systems toxicology impacting drug development." Current Opinion in Toxicology 23 (2020): 67-73. [2] Sové, Richard J., et al. "Virtual clinical trials of anti-PD-1 and anti-CTLA-4 immunotherapy in advanced hepatocellular carcinoma using a quantitative systems pharmacology model." Journal for immunotherapy of cancer 10.11 (2022): e005414. [3] Jeon, Miji, Hye-Won Kang, and Songon An. "A mathematical model for enzyme clustering in glucose metabolism." Scientific reports 8.1 (2018): 1-14 [4] Verma, BK., et al., "Model-based virtual patient analysis of human liver regeneration predicts critical perioperative factors controlling the dynamic mode of response to resection." BMC Systems Biology 13.1 (2019): 1-15. [5] Mulukutla, Bhanu Chandra, et al. "Bistability in glycolysis pathway as a physiological switch in energy metabolism." PloS one 9.6 (2014): e98756. [6] Kapuy, O., Makk-Merczel, K. and Szarka, A., 2021. Therapeutic approach of KRAS mutant Tumours by the combination of pharmacologic ascorbate and chloroquine. Biomolecules, 11(5), p.652.
Minor editing of English language required
Author Response
We thank the reviewer for these very important points and helpful suggestions. We have fixed the abbreviation and citation issues according to the reviewer’s advice. Moreover, we have added information about the first-line therapy of HCC (line 171-173). In addition, in the discussion section, we discussed that future studies may apply QSP models to examine the translation potential of PKM2 in CLDs (line 407-412).
Reviewer 3 Report
Dear Authors,
your manuscript is interesting and overall informative. Still some points need to be amended. Please find as an attachment the PDF of your manuscript with comments and suggestions. In addition, I think your manuscript would improve by adding one more Figure related to the first part.
Best regards

Dear Authors,
English Language needs to be improved, from both a grammar and syntax point of view.
Author Response
We thank the reviewer for these very important points and helpful suggestions. We have re-paraphrased the manuscript according to the advice given by the reviewer. The manuscript has been edited by Paperpal for grammar issue. Moreover, we have added a new Figure to the first part to describe the difference between PKM2 and other pyruvate kinase isoforms in metabolic reprogramming.

Round 2
Reviewer 3 Report
Dear Authors,
your manuscript has now improved.
I would just suggest you to add some recent References, which can enrich your manuscript. You can add them in paragraphs 1 and 3.
- doi: 10.3390/ijms24043710 PMID: 36835122
- doi: 10.1016/j.thromres.2020.12.002 PMID: 33340925
- doi: 10.1007/s00109-019-01780-2 PMID: 30953079